# *Beauveria bassiana* Ribotoxin (BbRib) Induces Silkworm Cell Apoptosis via Activating Ros Stress Response

Xiaoke Ma [1,2,†], Qi Ge [2,3,†], Rehab Hosny Taha [4], Keping Chen [2,*] and Yi Yuan [2,*]

1  The Laboratory Animal Research Center, Jiangsu University, Zhenjiang 212013, China; maxk123@163.com
2  School of Life Sciences, Jiangsu University, Zhenjiang 212013, China; geqi0616@163.com
3  School of the Environment and Safety Engineering, Jiangsu University, Zhenjiang 212013, China
4  Agricultural Research Center, Plant Protection Research Institute, Giza 8655, Egypt; Dr.rhhosny@gmail.com
*  Correspondence: kpchen@ujs.edu.cn (K.C.); yuanyi@ujs.edu.cn (Y.Y.)
†  These authors contributed equally to this work.

**Abstract:** The *BbRib* gene participates in the infection process of *Beauveria bassiana* (*B. bassiana*). It also helps pathogenic fungi to escape and defeat the insect host immune defense system by regulating the innate immune response. However, model insects are rarely used to study the mechanism of fungal ribosomal toxin protein. In this study, BbRib protein was produced by prokaryotic expression and injected into silkworm (*Bombyx mori*) larvae. The physiological and biochemical indexes of silkworm were monitored, and the pathological effects of BbRib protein on immune tissues of silkworm were examined by Hematoxylin and Eosin (HE) staining. BbRib protein can significantly affect the growth and development of the silkworm, causing poisoning, destroying the midgut and fat body and producing physiological changes. The ROS stress response in the adipose tissue and cells of the silkworm was activated to induce apoptosis. These results indicated that the *BbRib* gene not only participates in the infection process of *B. bassiana*, it also helps the pathogenic fungi escape the immune system by regulating the innate immune system of the silkworm, allowing it to break through the silkworm's immune defense. This study reveals the potential molecular mechanism of *BbRib* protein to insect toxicity, and provides a theoretical basis and material basis for the development and use of novel insecticidal toxins.

**Keywords:** *Beauveria bassiana* (*B. bassiana*); ribotoxin; silkworm; insecticidal actively; immune pathway



## 1. Introduction

Fungal pathogens are living entomopathogenic fungi (mainly spores) or their active components that can participate in the biological control of insect pests [1]. Many fungal insecticides can replace chemical insecticides. They have the characteristics of a wide host range, long residual activity and strong diffusion [2,3]. Most fungi are harmless to humans and animals and will not pollute the environment. The pathogenic mechanism of entomopathogenic fungi is usually controlled by multiple genes, which makes it difficult for pests to develop resistance. In addition, entomopathogenic fungi are the only microorganisms among the insect pathogens that can invade and infect insects through the epidermis; so, they have a unique advantage in the biological control of stinging–sucking pests [4].

Fungus-based insecticides that have been developed and applied to biological pest control currently mainly include *B. bassiana*, *Beauveria brongniartii*, *Metarhizium robberstii* and *Metarhizium acridum* [5]. When an entomopathogenic fungus penetrates the body wall of the host insect and enters the haemocoel, it needs to overcome the innate immune defense response of the host and then use the nutrients in the host body for growth and reproduction. These activities ultimately lead to the death of the host insect [6,7]. The types and levels of gene expression of pathogenic fungi vary greatly under the induction of secretions

such as insect body wall, hemolymph or plant root. Under a high-temperature (32 °C) or hyperosmotic environment (0.5-M NaCl) or insect body wall culture, the expression of the BCNA of the A subunit encoding gene CaN in *B. bassiana* was significantly increased by calcinulinase, and the expression level varied with the induction time [8]. Other studies have shown that the MaSte12 gene of *Metarhizium anisoplidae* affects conidial germination but does not affect fungal sporulation or conidial stress resistance. The MaSte12 gene affects the pathogenic ability of a locust by influencing appressorium differentiation. Transcriptome sequencing of the appressorium spores showed that the MASTE12 gene mainly controls the differentiation and penetration of appressorium fungi by regulating the expression of genes related to signal recognition, signal transduction, hydrophobic protein expression, fatty acid metabolism, carbohydrate metabolism, cell wall remoderase and other genes on the insect body surface, as well as the expression of somatic wall degrading enzymes in grasshoppers [9,10].

Ribosome-inactivating proteins (RIPs) in fungi are a type of ribonuclease (RNase) that can specifically modify the RNase on the large subunit of the ribosome by destroying the ribosome structural integrity and inhibiting protein biosynthesis [11,12]. Fungal ribotoxin is a type I RIP that can be secreted outside the cell, because it can enter the host cell and interact with the conserved sequence (sarcin–ricin loop, SRL) on the 28S rRNA on the large subunit of the ribosome of the cell. It can cause cleavage and destruction, which inhibit the translation and synthesis of proteins in the host cell, thereby causing a toxic effect on the cell [13,14]. The biological functions of fungal ribosomal toxins have not been fully elucidated, but many studies have confirmed that the fungal ribosomal toxin protein family has rich and diverse biological activities, the most important of which is ribosomal targeting. In addition, it is also involved in neurotoxicity, antibacterial activity, immunomodulatory activity, enzyme inhibitor activity and antiviral activity [15].

The function of the ribosomal toxin *BbRib* gene of *B. bassiana* has not yet been resolved, and its pathogenicity to insect hosts is unclear. The role of the toxin gene in the infection process of *B. bassiana* and the mode of action are unclarified. Therefore, detailed research on the pathogenic mechanism of entomopathogenic fungi may help to improve their insecticidal efficiency. An assessment of whether the insecticidal toxin gene can become a new biocontrol factor provides a basis for the development of novel insecticidal toxin preparations.

## 2. Materials and Methods

### 2.1. Prokaryotic Expression, Purification AMD Identification of Recombinant Target Protein

Pick the BL21 single clone that contains the recombinant expression plasmid, then inoculate it into 3 mL of Kana+antibiotic-containing LB broth overnight at 37 °C at 220 rpm. When the bacterial solution OD600 ≈ 0.6, add IPTG to a final concentration of 1 mM, continue to incubate for 4–5 h, then collect the bacterial solution into a 50-mL centrifuge tube, centrifuge at $4000 \times g$ at 4 °C for 20 min. Discard the supernatant, and collect the precipitate. After collecting the bacterial cells induced by IPTG by centrifugation, place them on ice and ultrasonically break them. Set the ultrasonic frequency to 200 W, 5 s ultrasound, 20 s interval, 30 cycles, and repeat three times at each repetition interval. Centrifuge at 4 °C and 12,000 rpm for 30–60 min, and collect the supernatant. Take a certain number of Ni-NTA Beads, add them to the column, and then equilibrate with 10 column volumes of 10-mM Tris (pH 7.4), let the Beads settle naturally during the column equilibration, and use a suitable ultrafiltration tube to purify the target protein. After detecting the protein concentration by Bradford method, it can be used in follow-up experiments and verified by SDS-PAGE and Western blot.

The target protein was identified by Shanghai Applied Protein Technology Co. Ltd. (Shanghai, China) using high-resolution liquid chromatography-mass spectrometry (Q Extractive plus, Thermofisher, Germany). The raw file of the mass spectrometry test was used to search the corresponding database by using the Proteome Discoverer1.4 software

(Thermo Fisher Scientific, Waltham, MA, USA), and finally the result of the identified protein was obtained.

### 2.2. The Establishment of a Blood Cavity Injection of BbRib Protein to Kill the Silkworm

The fifth-instar larva of the silkworm was used as the test insect; it was anesthetized on ice, and the surface of silkworm was wiped gently with 75% alcohol for disinfection. The BbRib protein was injected into the blood cavity of the test insect with a sterile syringe. The specific injection dose was dependent on the experiment. According to the preliminary experiments, the control group was treated with Phosphate Buffer Saline (PBS) buffer, and the injection group was reared under normal conditions. Then the physiological indicators of the silkworm (such as body color, feeding, weight, and activity status) were observed and recorded.

### 2.3. Preparation and Staining of Insect Paraffin Sections

The fifth-instar larvae of the normal silkworm infected with *B. bassiana* or as a control were fixed on the wax plate with the needle of a syringe, and the body was cut longitudinally along the dorsal blood vessel on the back of the silkworm with dissecting scissors. The syringe needle was used to separate the skin to both sides, the tissues or organs in the body were grasped using the tweezers, and then quickly rinsed in a sterile 0.85% NaCl solution to clean the hemolymph adhering to the surface of the tissue. Then the samples were fixed in the fixative overnight, which was then replaced with fresh fixative; they were put in 70% ethanol after 24 h, and stored at 4 °C. In addition, the tissue was embedded, the wax block was fixed on the sample holder of a Leica microtome (Leica RM2235, Leica Biosystem, Nussloch, German), and sliced manually. The thickness of the slice was adjusted according to the condition of the tissue sample, usually set to 5 μm. After deparaffinization, hematoxylin-eosin staining was performed. All slice samples were observed and photographed under a fluorescence microscope (Olympus BX51, Tokyo, Japan).

### 2.4. ROS Activity Level and Apoptosis Level in Insect Cells or Tissues

In a sterile 0.85% NaCl solution, sterilized forceps were used to dissect, and the silkworm tissues were collected for Reactive Oxygen Species (ROS) activity level detection. The H2DCF staining solution was prepared according to the operating instructions of the ROS assay kit (Beyotime, Shanghai). Then the solution was quickly added to the samples and incubated at 37 °C for 30 min in the dark. The H2DCF staining solution was aspirated with a pipette, and the tissues were quickly washed once with HBSS solution, and the tissues were transferred to a glass slide with holes (HBSS solution was added to the hole in advance), then directly observed under a fluorescence microscope. The Beyotime Annexin V-FITC cell apoptosis detection kit was used to detect the apoptosis level according to the instructions. The Annexin V-FITC/PI staining solution was prepared, and quickly added to the tissue before incubation at 25 °C for 20 min in the dark. Then the Annexin V-FITC/PI staining solution was aspirated with a pipette and washed with PBS buffer solution, and the tissues were transferred to a glass slide with holes (add PBS solution to the hole in advance), then directly observed under a fluorescence microscope.

### 2.5. Statistical Analysis

All data were plotted using GraphPad Prism 6. Student's *t* test was used to analyze and compare the significance of differences between groups. * $p < 0.05$, ** $p < 0.01$ and *** $p < 0.001$.

## 3. Results

### 3.1. Prokaryotic Expression, Purification and Mass Spectrometry Identification of BbRibProtein

To further determine the biochemical function and specific mechanism of BbRib protein, it was necessary to obtain functionally active BbRib purified protein through an in vitro protein expression and purification system. First, the *E. coli* prokaryotic expression

system was used to induce the expression and purification of BbRib protein in vitro. The constructed recombinant expression plasmid pET30a-BbRib was transformed into *E. coli* BL21 competent cells, expression was induced by IPTG, and the bacteria were collected by centrifugation. After sonication, the cell debris was removed by high-speed centrifugation and the supernatant was collected. After filtration and sterilization through a microporous membrane (0.22 μm), the supernatant was purified by a His-Ni column to collect the target protein, using Tris buffer containing different concentrations of imidazole gradient elution Ni column to simultaneously collect the target protein. To obtain a higher concentration of the target protein, we selected a 3-kDa ultrafiltration tube (Millipore) based on the predicted molecular weight of the BbRib protein (13.06 kDa) and concentrated the protein stock solution by ultrafiltration. The purified target protein BbRib was subjected to SDS-PAGE gel electrophoresis analysis and Western blot verification. The results showed that there was only a single and obvious band between 10 and 15 kDa in lane BbRib, and the position was in line with the expected molecular weight (13.06 kDa) of the target protein BbRib (Figure 1A). After detection with His-Tag primary antibody and DAB coloring, we observed that a single band was displayed at the same position on the PVDF membrane (Figure 1B), indicating that the prokaryotic expression and purification of BbRib protein in *E. coli* were successful.

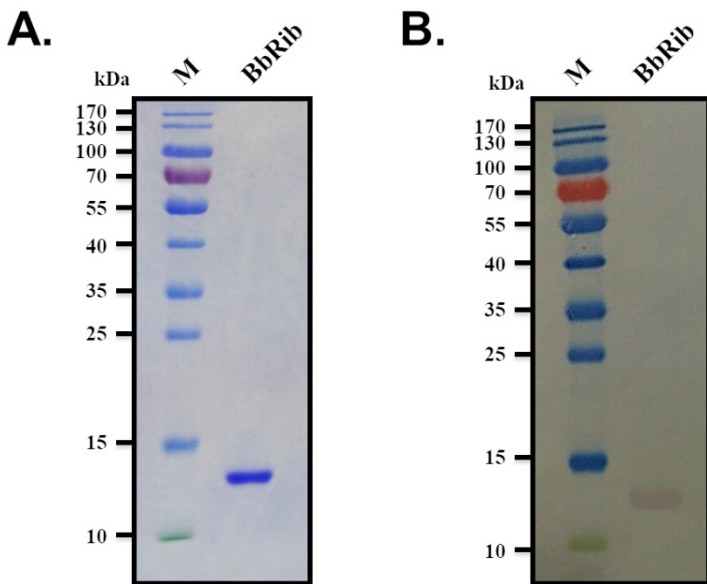

**Figure 1.** SDS-PAGE analysis and Western blot verification of purified BbRib. (**A**) SDS-PAGE analysis of purified BbRib protein. (**B**) Western blot verification. M, Prestained Protein Ladder (Thermo Fisher Scientific); Lane BbRib, purified BbRib protein.

### 3.2. Protein Profile Identification

To further verify the sequence correctness of the recombinant BbRib protein, we used a clean scalpel blade to tap the single protein band in the gel stained with Coomassie Brilliant Blue according to the identification results of SDS-PAGE gel electrophoresis analysis and Western blot. In addition, we entrusted Shanghai Applied Protein Technology Co. Ltd. (Shanghai, China) to perform mass spectrometric identification of the samples. MALDI-TOF was used to analyze the proteolysis products by MS1 and MS2 mass spectrometry. The qualitative information of the protein was finally obtained by collecting peptide molecules and their fragments and then matching the database software. Based on the identification results provided by APT, the purified BbRib protein was identical to the wild-type *B. bassiana* (*Beauveria bassiana* ARSEF 2860) Ribonuclease/Ribotoxin (XP_008594335.1) amino acid sequence provided by NCBI, and the identification is correct (Figure 2).

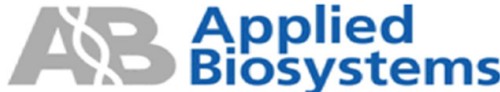

**Analysis Information**

| | | | |
|---|---|---|---|
| Report Type | Protein-Peptide Summary by Spot | Analysis Type | Combined (MS+MS/MS) |
| Sample Set Name | Sample project20170921 | Database | NCBI_Beauveria |
| Analysis Name | P20170900559 | Creation Date | 10/11/2017 12:53:37 |
| Reported By | 10/11/2017 13:53:34 - admin | Last Modified | 10/11/2017 12:54:43 |
| MS Acq. : Proc. Methods | (Unspecified) : (Unspecified) | | |
| Interpretation Method | (Unspecified) | | |

| Gel Idx/Pos | 193/H20 | | | Instr./Gel Origin | | BA2151/Sample Project 20170921 | | Process Status | | Analysis Succeeded | |
|---|---|---|---|---|---|---|---|---|---|---|---|
| Plate [#] Name | [1] Sample Project 20170921 | | | Instrument Sample Name | | | | Spectra | | 11 | |

| Rank | Protein Name | Species | Accession No. | Protein MW | Protein PI | Pep. Count | Protein Score | Protein Score C. I. % | Total Ion Score | Total C. |
|---|---|---|---|---|---|---|---|---|---|---|
| 1 | Ribonuclease/ribotoxin [Beauveria bassiana ARSEF 2860] | | XP_008594335.1 | 13047.2 | 7.66 | 6 | 383 | 100 | 341 | |

**Protein Group**

| | | | | |
|---|---|---|---|---|
| Ribonuclease/ribotoxin [Beauveria bassiana ARSEF 2860] | | EJP70147.1 | 13047.2 | 7.6599 998474 1211 |

**Peptide Information**

| Calc. Mass | Obsrv. Mass | ± da | ± ppm | Start Seq. | End Seq. | Sequence | Ion Score | C. I. % | Modification | Rank | Result Type |
|---|---|---|---|---|---|---|---|---|---|---|---|
| 1249.5808 | 1249.5979 | 0.0171 | 14 | 52 | 64 | SGGTAGSSTYPHK | | | | | Mascot |
| 1249.5808 | 1249.5979 | 0.0171 | 14 | 52 | 64 | SGGTAGSSTYPHK | 95 | 100 | | | Mascot |
| 1280.5867 | 1280.6028 | 0.0161 | 13 | 86 | 99 | SGGVYTGGSPGADR | | | | | Mascot |
| 1280.5867 | 1280.6028 | 0.0161 | 13 | 86 | 99 | SGGVYTGGSPGADR | 108 | 100 | | | Mascot |
| 2545.1711 | 2545.1833 | 0.0122 | 5 | 65 | 85 | YNNYEGFNFPVSGPYYEFPIK | | | | | Mascot |
| 2545.1711 | 2545.1833 | 0.0122 | 5 | 65 | 85 | YNNYEGFNFPVSGPYYEFPIK | 138 | 100 | | | Mascot |
| 2897.3193 | 2897.3352 | 0.0159 | 5 | 25 | 51 | AATTCGTVYYTANQVNAASQAACNYVK | | | Carbamidomethyl (C)[5,23] | | Mascot |
| 3775.7341 | 3775.8186 | 0.0845 | 22 | 52 | 85 | SGGTAGSSTYPHKYNNYEGFNFPVSGPYYEFPIK | | | | | Mascot |
| 3806.7397 | 3806.8599 | 0.1202 | 32 | 65 | 99 | YNNYEGFNFPVSGPYYEFPIKSGGVYTGGSPGADR | | | | | Mascot |

| Database | Number of seuences | Date of fasta file | dye method |
|---|---|---|---|
| NCBI_Beauveria_bassiana | 33990 | 10/11/2017 | Blue |

**Figure 2.** Identification of purified recombinant BbRib by MALDI-TOF mass spectrometry. Purified recombinant protein BbRib was analyzed and identified successfully by MALDI-TOF mass spectrometry. Amino acid sequence alignment with *Beauveria bassiana* ARSEF 2860 Ribonuclease/Ribotoxin (XP_008594335.1) in the NCBI database confirmed the verification.

*3.3. Changes of Physiological Indexes and Histopathological Phases of the Silkworm after BbRib Injection into the Haemocoel*

Purified BbRib protein (25 μg/larva) was injected into the blood cavity of silkworm larvae, and then, physiological indicators (such as body color, feeding, weight, and activity status) of the larvae were continuously monitored. Moreover, according to the external poisoning symptoms exhibited by the test silkworms, anatomical sampling and pathological examination of tissue sections were carried out in a timely manner, and based on this, the potential toxic mechanism of BbRib protein was analyzed and speculated. As shown in Figure 3A,B, after the BbRib protein was injected into the haemocoel of the silkworm larvae, the weights of the test larvae declined from the onset of poisoning symptoms to the death of the larvae at 24 h. Compared with the control group that was fed normally and injected with PBS buffer, the silkworm larvae in the BbRib protein treatment group showed symptoms of poisoning such as coma, refusal to feed, and lying down at 6 h after injection, and the larval weight was significantly reduced. With the prolongation of the postinjection time (12 and 24 h), the symptoms of poisoning intensified. The activity of the test insects declined, and the weights continued to decrease. Compared with the control silkworm larvae, the difference in the average weight between the groups was more significantly increased.

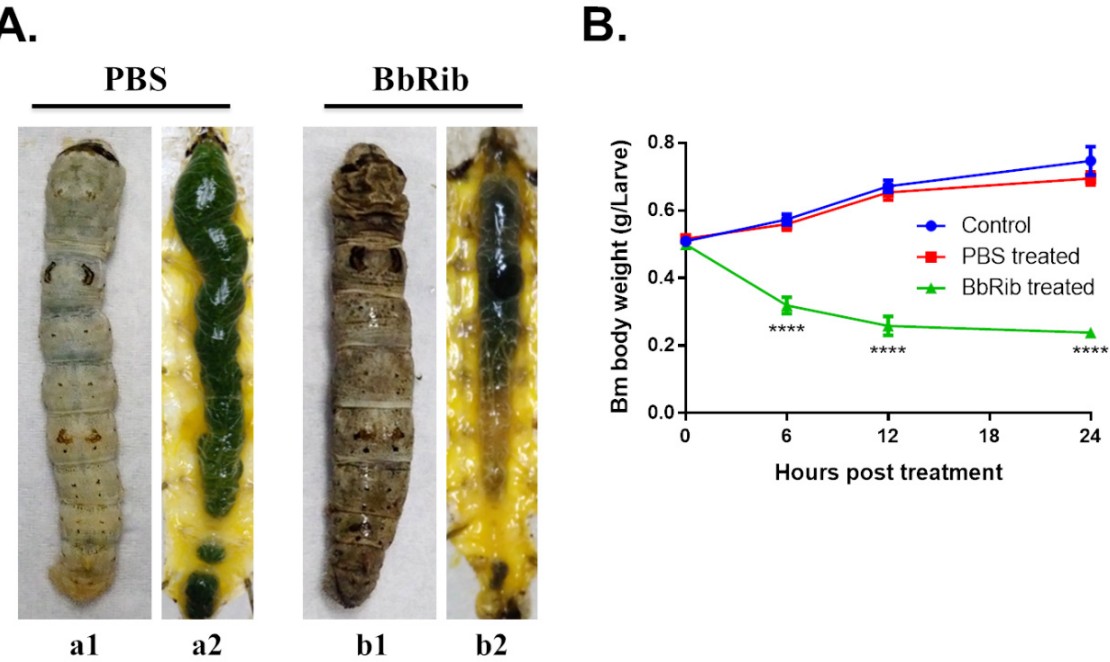

**Figure 3.** Morphological changes of silkworms after BbRib treatment via hemocoel injection. (**A**) Silkworm larvae were treated with purified BbRib (25 μg/larva) or PBS buffer (control group). Dead insects were removed and dissected to observe tissues and organs in the haemocoel. a1 and a2, morphological changes and pathological anatomy of PBS buffer control group; b1 and b2, morphological changes and pathological anatomy of BbRib protein treatment group. (**B**) Effect of BbRib on the weight of silkworm larvae after haemocoel injection. Average body weights of fifth-instar silkworm larvae were measured and recorded after BbRib treatment and compared with PBS control group at 6, 12 and 24 h, respectively. ****, $p < 0.0001$.

At 12 h after the injection of BbRib protein, the silkworm larvae continued to die. Therefore, we picked out the dead and captured photographs while dissecting and observing them (Figure 3A). The silkworm test larvae were injected with BbRib protein (25 μg/larva) through the blood cavity to the stage of death (12 h), compared with the control group silkworm treated with PBS buffer (Figure 3A). The surface cuticle of the treatment larvae was blackened; the head skin folds increased, and the internal depression was great; the middle part of the larval body (somites from the thorax to the abdomen) appeared swollen, and the body wall was moist compared to the control larvae, while the tail shrank (Figure 3A). To further understand the specific changes in the internal structure of the larvae, we dissected the test larvae. We found that the stimulation or treatment with BbRib protein produced serious damage, and damage to many tissues and organs occurred in the hemolymph system, leading to serious physiological changes. In the process of dissecting the larvae for testing, with the passage of time, obvious pathological changes occurred in the intestinal and surrounding fat body tissues. First, the upper part of the midgut of the larvae was atrophied, and almost no food residues were found in the foregut region from the oral cavity through the throat to the esophagus (Figure 3A). These observations showed that BbRib protein can trigger an antifeedant effect, and this can also be used as the larval body weight. This is one of the possible reasons for the significant weight reduction (Figure 3B). Second, the midgut contents of the test insects were severely blackened, the surface structure of the intestinal tissue was dissolved and ulcerated and turned black, a large amount of exudation of the contents was observed, the length of the intestine shortened, the volume of the intestinal cavity was reduced, and there was no food in the rear portion of the midgut. The BbRib protein produced the greatest damage to the hindgut. Compared with the PBS buffer control group, the H1 and H2 structures of the hindgut were clear and complete. The distinguishing color was normal, and almost no hindgut tissue was observed in the BbRib protein treatment group.

### 3.4. Effect of BbRib Protein on Pathological Changes of Silkworm

We initially observed that the BbRib protein was toxic and lethal to silkworm larvae. To further explore the toxic mechanism of BbRib, we tested the important tissues of silkworm, such as the midgut, after haemocoel injection of BbRib protein. The pathological phases of the midgut, hindgut and fat body were continuously observed and compared under a microscope using tissue sections and HE staining methods (Figure 4). Our results showed that the midgut tissue structure of the larvae in the PBS buffer control group was complete, with developed epithelial cells. Columnar cells, cupped cells and regenerative cells were clearly distinguishable under the microscope after HE staining. The cell morphology and size were normal. The surface of the intestinal wall had several folds; samples were taken at different time points for tissue sectioning and staining observation, and no significant difference was observed. The pathological changes of the intestinal tissue of silkworm larvae treated with BbRib protein were obvious. At 6 h after the injection of BbRib protein, the columnar nucleus in the midgut epithelial cells began to expand, the cup vesicle structure inside the goblet cells was destroyed, the folding structure of the midgut became loose, but the periphagus membrane was still clearly visible. At 12 h after injection of BbRib protein, the columnar epithelial cells of the midgut decreased, but the degree of nucleus enlargement increased. The cup vesicle structure inside the goblet cells was further destroyed; the folds of the intestinal wall disappeared, and the microvilli layer was significantly thickened. We speculate that BbRib protein had completely penetrated into the intestinal lumen at this time, and it is transparent to some components in the periphagus. At 24 h after the injection of BbRib protein, the enlarged nucleus of the columnar cells in the midgut was squeezed out and continued to fall off into the intestinal lumen.

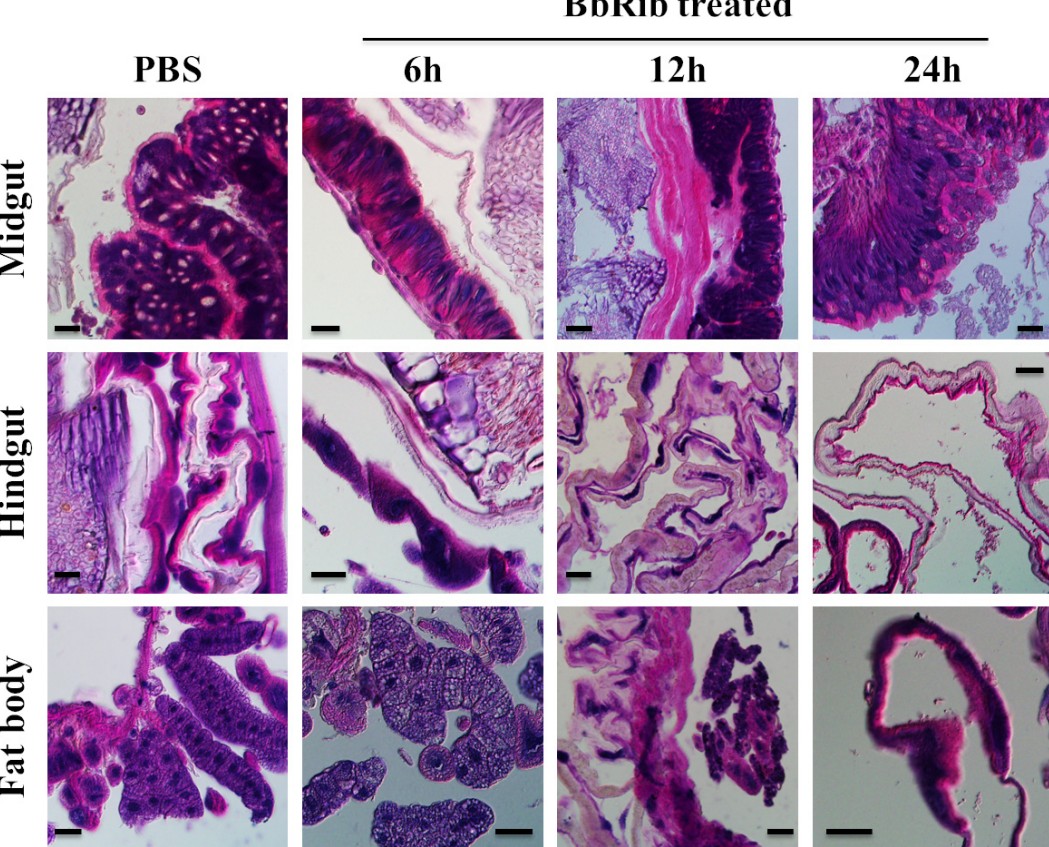

**Figure 4.** Histopathological changes and microscopic observation of silkworm larvae after BbRib treatment. Silkworm larvae were treated with purified BbRib (25 μg/larva) or PBS buffer (control group). Silkworm tissues (midgut, hindgut and fat body) were sampled using paraffin sections and subjected to hematoxylin-eosin (HE) staining. The histopathological changes of samples were compared and analyzed by microscopic observation.

The hindgut tissue structure of silkworm larvae in the PBS control group was complete. The various layers of the intestinal wall tissue such as the muscle layer, base membrane, epithelial cell layer and intima were clearly distinguishable under the microscope, and the intestinal cavity was filled with food (fresh mulberry leaves). The pathological changes in the hindgut tissue of the larvae in the BbRib protein treatment group were obvious. At 6 h after injection of BbRib protein, the thickness of the hindgut intestinal wall tissue was uneven, and the gap between the layers increased. Some food and residues were still wrapped in the intestinal cavity. At 12 h after injection of the BbRib protein, no food or residue was found in the hindgut intestinal cavity. At 24 h after the injection of BbRib protein, the tissue in the intestinal cavity continued to ablate, and the gap between each level was larger.

We also sampled and observed the fat bodies of silkworm larvae at different onset periods. The structure of the fat body of the larvae in the PBS buffer control group was clear and discernible, mainly distributed in the periphery of the digestive tract and other organs, with a regular spindle shape. The pathological changes of the fat body tissue of larvae treated with the BbRib protein were very obvious. Six hours after the injection of BbRib protein, part of the basal layer of fat body cells was broken; the cells were lysed into free single cells, the nucleus fell off, and the cell contents were released into the haemocoel. At 12 h after the injection of BbRib protein, the fat body cells became significantly smaller and gradually stretched from a spindle shape to a slender shape or a ribbon shape. At 24 h after the injection of BbRib protein, all the silkworm larvae had died. At this time, the fat body cells were completely ablated, and no complete and discernible fat body structure could be observed under the microscope.

### 3.5. Stress Changes of ROS Activity in the Fat Body of Silkworm after Haemocoel Injection of BbRib

We also dissected silkworm larvae with BbRib protein injected into the haemocoel at different time points (6, 12 and 24 h) and analyzed the level of reactive oxygen species (ROS) in the fat body tissue (Figure 5). If sampling is performed 6 h after injection of BbRib protein, a strong green fluorescence signal was observed under a fluorescence microscope compared with the PBS control group. At 12 h after injection, the fluorescence intensity of the BbRib protein treatment group was weaker than the 6-h result, but it was still significantly higher than that of the control group. At 24 h after the injection, the BbRib protein treatment group was similar to the control group, and almost no green fluorescence was detected. These results indicate that the purified BbRib protein can quickly activate the ROS stress response in the fat body tissues and cells of the larvae after being injected into the blood cavity, but its activity gradually decreases as the injection time increases, and the degree of insect poisoning increases. We speculate that the ROS oxidative stress response is mainly involved in the detoxification process in the body during the early stage of poisoning, but the effect is not obvious in the later stages.

### 3.6. Injecting BbRib into the Haemocoel Can Induce Apoptosis of Adipose Tissue in the Silkworm

While detecting the level of ROS in the fat body tissue of the silkworm larvae, we also performed apoptosis analysis on the sampled fat body tissue (Figure 6). In contrast to the changes of ROS activity levels, after BbRib protein was injected into the haemocoel of larvae, compared with the negative control group treated with PBS, the apoptotic level of the fat body tissue was prolonged with the injection time. The degree of poisoning increased. These results indicate that, with the accumulation of BbRib protein's action time and toxicity in the larvae, the degree of apoptosis and necrosis of the insect fat body tissue increased and eventually led to tissue disintegration and death.

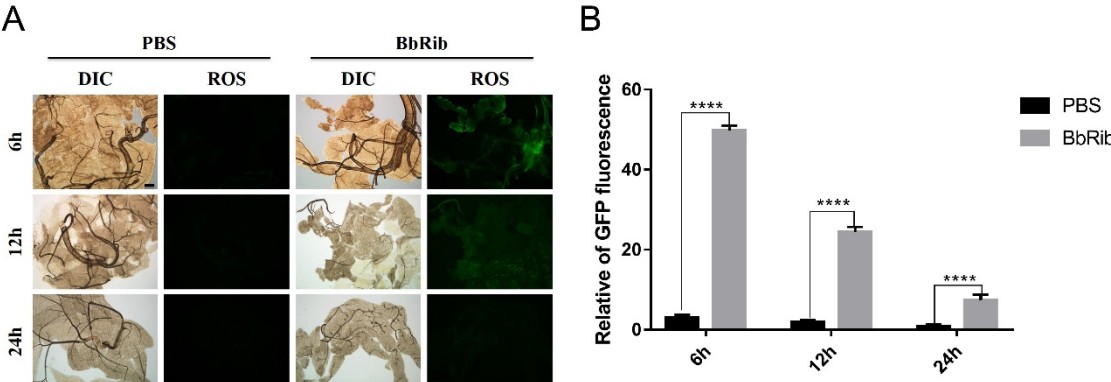

**Figure 5.** BbRib induced change of ROS activity level of silkworm fat bodies via haemocoel injection. BbRib (25 μg/larva) was injected into larvae via haemocoel injection. (**A**) At different time points (6, 12 and 24 h), fat bodies were sampled and subjected to a ROS activity assay using a fluorochrome staining method under microscopic observation. (**B**) The quantified ROS level at different time points. Continuous dynamic observation and comparisons were carried out between different groups (PBS buffer was the negative control) for the analysis and evaluation of its influence on the ROS activity level of insect fat bodies. ****, $p < 0.0001$.

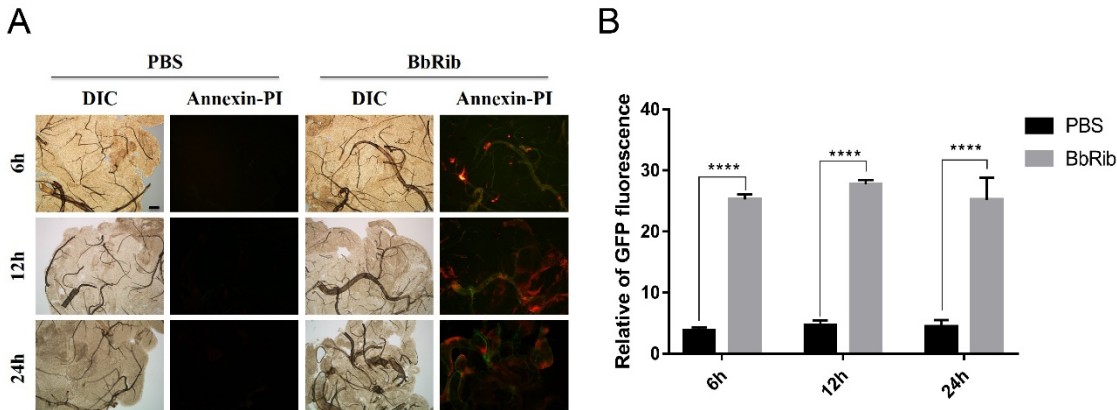

**Figure 6.** BbRib induced cellular apoptosis of silkworm fat bodies by haemocoel injection. BbRib (25 μg/larva) was injected into silkworm larvae via haemocoel injection. (**A**) At different time points (6, 12 and 24 h), fat bodies were sampled and subjected to cellular apoptosis assay using Annexin V-FITC/PI staining method under microscopic observation. (**B**) The quantified GFP fluorescence level at different time points. Continuous observation and comparisons were made between different treatments groups (PBS buffer was the negative control) for the analysis and evaluation of its influence on the apoptosis level of insect fat bodies. ****, $p < 0.0001$.

## 4. Discussion

Chemical pesticides have traditionally been the most important tools for controlling agricultural and forestry pests. However, due to their long-term excessive use, chemical pesticides have caused ecological problems such as pest resistance and environmental pollution [16]. With the development of a social economy and increased awareness of environmental protection, the development and application of environmentally friendly biological pesticides have received more attention [17]. Biological materials are important sources for the development of new pesticides. The research and development of insect pathogenic fungi, bacteria or viruses for microbial pesticides have a historical basis. Among these, fungal insecticides have good safety, persistent control effects and easy artificial cultivation. Fungi are ideal choices for the development of new biopesticides [18].

Similar to most plant pathogenic fungi, entomopathogenic fungi need to undergo appressorium differentiation when penetrating insect body walls. They rely on the combined action of mechanical pressure formed by appressorium and degrading enzymes. Therefore,

when entomopathogenic fungi invade the insect body wall, they secrete a large number of extracellular hydrolases, such as protease, chitinase, and lipase, to help the fungus invade. When the entomopathogenic fungus penetrates the insect body wall and enters the hemolymph. After entering the haemocoel, it is necessary to overcome the host's innate immune defense response first and then effectively obtain the nutrients in the host for growth and reproduction. This ultimately leads to the death of the host insect [7,19]. Pathogenic fungi are induced by the secretions of insect body walls, hemolymph or plant roots, and there are great differences in the types and levels of gene expression. For example, in *B. bassiana*, under higher temperature (32 °C) or a hypertonic environment (0.5-M NaCl) or insect body wall cultures, the calcineurin catalyzes the homology of the A subunit encoding gene CAN. The expression level of the source gene (BCNA) will increase significantly, and the change trend of the expression level with the induction time is different [20]. At different stages of infection, entomopathogenic fungi will adopt completely different genetic strategies. For example, in the process of body wall infection, the formation of appressorium and related hydrolase usually play an important role, and the reassembly of cell wall structure and the secretion efficiency of insecticidal toxins determine whether the pathogenic fungus can be contained in the host.

Insects have an innate immune defense system, and the recognition of foreign pathogens is the first step in initiating the downstream immune response. This process is mainly mediated by different pattern recognition proteins. Currently discovered and identified insect pattern recognition proteins mainly include peptidoglycan recognition proteins (PGRPs), immunoglobulin-like proteins, β-1,3-glucan binding proteins, C-type lectin and multifunctional apolipoproteins [21]. However, corresponding to insect immune recognition, insect pathogenic microorganisms have also co-evolved immune evasion ability with molecular diversity. Therefore, in order for entomopathogenic fungi to successfully colonize the haemocoel of host insects, different immune evasion strategies must be adopted at different stages of infection to avoid the rapid activation of insect blood cells such as phagocytes, cysts, and black spots. Innate immune responses include the expression and secretion of antimicrobial peptides [22,23].

Many important virulence and efficacy genes have also been discovered in *B. bassiana*. For example, the Bbslt2 gene, which encodes mitogen-activated protein kinases (MAPKs), can significantly inhibit the growth and spore production of fungi, and it is an indicator of cell wall damage. Congo red, and fungal cell wall degrading enzymes. Sensitivity increased, while the insecticidal virulence decreased significantly [24]. In another example, the chitinase-encoding gene Bbchit1 was overexpressed in *B. bassiana*; the half-lethal dose of the transformant to aphids was significantly reduced, and the insecticidal virulence was significantly increased [25]. Another example involves the non-ribosomal peptide synthetase (NRPS)-encoding genes bbBeas and bbBsls, which are responsible for the synthesis of beauverin and type II beauverin, respectively. They are both colonized by fungi in the host. In addition to the above-mentioned genes, the virulence effect genes found in *B. bassiana* include the surface hydrophobin coding gene hyd, the cytochrome P450 gene Bbcyp52x1, the neuronal calcium sensor gene Bbcsa1 and the G protein coupled receptor coding gene BbGPCR3. These genes can be used as virulence factors participating in host infection and affecting the insecticidal virulence of *B. bassiana* [26–29].

Among them, RIPs are a type of RNase that can specifically modify the ribonucleic acid rRNA on the large subunit of the ribosome by destroying the structural integrity of the ribosome and inhibiting protein biosynthesis [30,31]. RIPs are divided into two types according to their primary structure and enzymatic mechanism of action: type I RIPs and type II RIPs. Among them, type I RIPs are composed of a single polypeptide chain, which is a single-chain protein with RNase cleavage activity, represented by α-Sarcin and Saporin; type II RIPs have two polypeptide chains connected by disulfide bonds, and the A chain has I for the enzymatic activity of type RIPs, while the B chain can bind glycoproteins, represented by Ricin and Shiga type toxins [32]. Among the nearly 7000 ribonucleotides in the ribosomes of eukaryotes, RIPs can destroy the integrity of the ribosome structure and

completely inactivate it by removing an adenine (A) or cutting off a single phosphodiester bond. Fungal ribotoxin is a type I RIP that can be secreted outside the cell, because it can enter the host cell and interact with the conserved sequence (SRL) on the 28S rRNA on the large subunit of the ribosome. Cut and destroy to inhibit the translation and synthesis of proteins in the host cell, thus having a highly toxic effect on the cell [13,33].

Generally speaking, the production of ROS in insect bodies is a crucial response of innate immune cells to pathogen infection, because the ROS can resist the infection of pathogens and play an antibacterial role [34–36]. Under normal conditions, the ability of the body's cells to produce ROS and detoxify ROS is balanced [37]. However, this balance could be disrupted by the infection of fungus, and more ROS would be produced in cells in response to fungus. In this study, the oxidative stress induced by BbRib toxins would generate ROS radicals that cause damage to silkworm fat bodies. Furthermore, the catalase is considered to be the only one that responsible for eliminating ROS. Ge et al. indicated that the pathogenic bacteria can damage the normal immune function of fat body cells, and also affect the antioxidant enzyme activities and ROS levels in silkworm [38]. Therefore, we speculate that the BbRib toxins can inhibit the activity of catalase, thus leading to the body's continuous production of ROS, which plays an anti-pathogenic role.

Although the biological functions of fungal ribosomal toxins have not yet been fully elucidated, many studies have confirmed that the identified fungal ribosomal toxin protein family has rich and diverse biological activities, the most important of which is still ribosome targeting. In addition, it is also widely involved in neurotoxicity, antibacterial activity, immunomodulatory activity, enzyme inhibitor activity and antiviral activity [39]. It is this characteristic of molecular and functional diversity that enables fungal ribosomal toxins to become research tools and probes in molecular biology, as well as 'rich areas' for the discovery of innovative drugs and pesticides. In addition, detailed research and analysis of the structure and function of the members of the fungal ribosomal toxin family lay the foundation for their use in the biological control of pests or the development of anti-tumor agents for the treatment of humans [40].

In summary, in this study, *B. bassiana* and the model insect *B. mori* were used as the research objects. We established a pathogenic fungus and a model insect infection interaction model using molecular biology, biochemistry, histopathology and other technical methods to conduct in-depth research on the biological function, mechanism of action and insecticidal activity mechanism of the *BbRib* gene. This approach revealed its mode of regulating the host insect immune defense response during pathogenic fungi infection. It confirmed that the *BbRib* gene not only participates in the infection process of *B. bassiana* but also assists the pathogenic fungus in immune evasion and finally in breaking through the host's immune defense by regulating the innate immune system of the insect host. We determined the effect of the *BbRib* gene on the growth and development, virulence and pathogenicity of *B. bassiana* and revealed the potential poisoning mechanism of BbRib protein on insects from molecular, cellular, tissue and individual levels. We therefore comprehensively evaluated the insecticidal toxin gene, which may lead to the development of a new generation of biocontrol factors.

**Author Contributions:** X.M. Writing—original draft preparation, investigation, data curation; Q.G. and R.H.T. resources, investigation, data curation; K.C. and Y.Y. Supervision, project administration, funding acquisition. All authors have read and agreed to the published version of the manuscript.

**Funding:** This work was supported by the National Natural Science Foundation of China (grant numbers 31900359, 31861143051 and 31872425).

**Institutional Review Board Statement:** Not applicable.

**Informed Consent Statement:** Not applicable.

**Data Availability Statement:** The datasets during and/or analyzed during the current study available from the corresponding author on reasonable request.

**Acknowledgments:** We also thank LetPub (www.letpub.com) accessed on 15 July 2021 for its linguistic assistance during the preparation of this manuscript.

**Conflicts of Interest:** The authors declare that they have no known competing financial interests or personal relationships that could have appeared to influence the work reported in this paper.

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
