# Peer review of "Beauveria bassiana Ribotoxin (BbRib) Induces Silkworm Cell Apoptosis via Activating Ros Stress Response"

_processes, doi:10.3390/pr9081470_

Round 1

Reviewer 1 Report

This study analyzed physiological function of BbRib in host insect. BbRib affects Bombyx larval development and increased ROS-induced cell death induction rate. There are several potentially interesting observations in the manuscript by Chen and coworkers. The research is well planned and the text is logically described. After revision, this manuscript can be published in Processes.

  1. There are 6 figures (Fig1~Fig6), but there are 7 figure legends (Fig1~Fig7????) in this paper. This point must be corrected before publication.
  2. Please present the ROS level in other tissue e.g. midgut, hindgut, and compere the quantified data of ROS level e.g. using Image J.
  3. Please present the data whether co-injection of RbRib and antioxidant (ex. Ascorbic acid, N-acetyl cysteine) decreases cellular ROS level and ROS-induced apoptosis.
  4. How did BbRib induce ROS production?

Author Response

Reviewer 1

This study analyzed physiological function of BbRib in host insect. BbRib affects Bombyx larval development and increased ROS-induced cell death induction rate. There are several potentially interesting observations in the manuscript by Chen and coworkers. The research is well planned and the text is logically described. After revision, this manuscript can be published in Processes.

  1. There are 6 figures (Fig1~Fig6), but there are 7 figure legends (Fig1~Fig7????) in this paper. This point must be corrected before publication.

Response: Thank you for your careful review. We have made corrections in our revised manuscript.

  1. Please present the ROS level in other tissue e.g. midgut, hindgut, and compere the quantified data of ROS level e.g. using Image J.

Response: Thank you. We have added the quantified data of ROS level in fat body tissues (Fig. 5B) according to your suggestion. However, since the infection is intravascular, the fat body is the first line of defense and a crucial innate immune organ. Therefore, in our research, we only detected the ROS levels in fat body tissues. Thanks again for your valuable comment.

  1. Please present the data whether co-injection of RbRib and antioxidant (ex. Ascorbic acid, N-acetyl cysteine) decreases cellular ROS level and ROS-induced apoptosis.

Response: Thank you very much for your professional suggestion in the areas of immune research and control design. We all very agree with your suggestion, if a parallel control (+/- antioxidant treatment) was set up during the experiment, the results will be more helpful in drawing our conclusions. However, we have already sampled the injected silkworm tissue and sent it to Shanghai Majorbio Bio-Pharm Technology Co., Ltd. for RNA-seq. We plan to systematically study the changes in ROS-related pathways based on the transcriptome data. At that time, we will use your recommended experimental protocol. In this study, please forgive us for not being so comprehensive. If you have any other queries, please do not hesitate to contact us.

  1. How did BbRib induce ROS production?

Response: Thank you for your valuable comment, your comment is of great help for improving our paper. Based on current preliminary phenotypic data and observation results, it is not yet possible to build a complete mechanism model. However, in order to give a comprehensive and in depth explanation of this issue, we are carrying out the RNA-seq analysis and hope to systematically explore the transcriptional level regulation of differently expressed gene involved in ROS-related pathways based on transcriptome data. For this issue, we have made some states in the discussion section. Please check it in our revised manuscript.

Reviewer 2 Report

The Xiaoke Ma et al. Manuscript is well designed and presented. Some questions and suggestions
1. Was the project approved by an investigation committee? You can add the approval number and the name of the committee. Thanks
2. I suggest adding the figures in the results section
3. Western blot studies, were they performed in triplicate? can you put the images?
4 were the results of the Western blot quantified?
5. in the results of Stress changes of ROS activity in the fat body of silkworm after haemocoel injection of BbRib. were free radicals quantified? or oxidative stress? or ROS?

Author Response

Reviewer 2

The Xiaoke Ma et al. Manuscript is well designed and presented. Some questions and suggestions

  1. Was the project approved by an investigation committee? You can add the approval number and the name of the committee. Thanks

Response: Thank you for your valuable comment, your comment is of great help for improving our paper. Actually, as the silkworm (Bombyx mori) is an invertebrate insect, there no ethics statement required. Thank you again for your suggestion.

  1. I suggest adding the figures in the results section.

Response: Thank you for your careful review. We have added the figures in the results section according to your suggestion.

  1. Western blot studies, were they performed in triplicate? can you put the images?

Response: Thank you for your valuable comment. Your comment is of great help for improving our paper. Actually, in our research, Western blot is only used for preliminary qualitative identification of whether the purification target protein we obtained is BbRib. As described in the part of Materials and Methods, the purified BbRib protein was sent to Shanghai Applied Protein Technology Co. Ltd. for mass spectrometry identification. We think that this result can more accurately confirm the correctness of the target protein. Thanks again for your comment.

4 were the results of the Western blot quantified?

Response: Thank you for your valuable comment. Your comment is of great help for improving our paper. Same as comment 3, Western blot is only used for preliminary qualitative identification of whether the purification target protein we obtained is BbRib. Therefore, the results of the Western blot do not need to be quantified. We all appreciate for your valuable comment.

  1. in the results of Stress changes of ROS activity in the fat body of silkworm after haemocoel injection of BbRib. were free radicals quantified? or oxidative stress? or ROS?

Response: Thank you for your valuable comment. We have added the quantified data of ROS level in fat body tissues (Fig. 5B) according to your suggestion. Actually, the ROS activity level in silkworm fat body were measured by using the ROS assay kit (Beyotime, Shanghai) which can use the fluorescent probe DCFH-DA for ROS detection. DCFH-DA itself has no fluorescence and can freely pass through the cell membrane. After entering the cell, it can be hydrolyzed by intracellular esterase to produce DCFH. DCFH cannot penetrate the cell membrane, so that the probe can be easily loaded into the cell. Then the ROS in the cell can oxidize non-fluorescent DCFH to produce fluorescent DCF. Therefore, we could detect the fluorescence of DCF to know the ROS levels in the fat body of silkworm. If you have any other queries, please do not hesitate to contact us.
